# Promoting healthy lifestyles among nurse trainees: Perceptions on enablers and barriers to dietary and physical activity behaviours

**Phidelia Theresa Doegah****\*, Evelyn Acquah\***

Institute of Health Research, University of Health and Allied Sciences (UHAS), Ho, Volta Region, Ghana

\* tdoegah@uhas.edu.gh, phideliadoegah@yahoo.com

## Abstract

### Introduction

Promoting healthy lifestyles is important to protect against obesity and non-communicable diseases. However, there is a lack of understanding of the enablers and barriers to healthy lifestyles (dietary and physical activity) among pre-service nursing trainees in Ghana. This study therefore aims to examine the perceived practice, enablers and barriers in adopting healthy dietary and physical activity behaviours.

### Materials and methods

Cross-sectional qualitative, one-on-one in-depth interviews with a semi-structured guide were conducted with 16 nursing trainees (public health nursing) (aged: 18–25 years) in the School of Nursing and Midwifery, University of Health and Allied Sciences. Participants were selected based on body mass index (underweight, normal, over-weight, and obesity) classification of the world health organization. The interviews were audio-recorded and transcribed. Data analysis was manually carried out.

### Results

Enablers and barriers were grouped into levels of individual (intrapersonal), social environment, physical environment, and university factors based on ecological model initially formulated by Bronfenbrenner's. Enablers to healthy dietary behaviour were self-discipline, dietary knowledge, social support, and access/availability. Enablers related to physical activity mentioned were body image, social support, and the existence of student societies. Barriers to healthy dietary behaviour included upbringing, preference, accessibility, safety/appearance, and studies/lectures. Barriers mentioned in relation to physical activity include a busy lifestyle, inadequate feeding, studies/academic activity, student societies, upbringing, and social support.

**Data Availability Statement:** There are legal restrictions on sharing the de-identified data. However, data would be provided upon reasonable request from the UHAS ethics committee. Contact:

Mr. Fidelis Anumu, Research Ethics Committee administrator, email: rec@uhas.edu.gh.

**Funding:** PTD Ghana Studies Association http://ghanastudies.org No: The funders had no role in the study design, data collection and analysis, decision to publish, or preparation of the manuscript.

**Competing interests:** The authors have declared that no competing interest exist.

## Conclusion

A program to support healthy lifestyles for nursing trainees is needed. Specifically, in terms of developing and implementing interventions to overcome barriers and promote facilitators to adopt healthy dietary and physical activity behaviours whilst in training.

## Introduction

Globally, modifiable lifestyle behaviours such as poor dietary behaviour, and physical inactivity, are risk factors for obesity and Non-communicable diseases (NCDs). Specifically, excess salt use, and physical inactivity have contributed respectively to 4.1 million, and 1.6 million deaths annually [1].

NCDs have been observed to occur at a late age after prolonged exposure to unhealthy lifestyles. However, it's recently been reported among the youth in low-and middle-income countries [2]. In Ghana, poor modifiable lifestyles, for instance, limited fruit and vegetable intake [3, 4], physical inactivity [5, 6] have been observed among the youth as well as the adult population. Unhealthy behaviours among young people including University students, although thought of as temporary, once acquired at this age, persists to adulthood [7]. Because it is a period for acquiring and shaping lifestyles.

The Ministry of Health (MoH) in 2005 adopted a Regenerative Health and Nutrition Programme (RHNP) to promote a healthy lifestyle agenda. This programme feeds into the global initiative of the World Health Organization (WHO) to address the increasing NCDs and obesity pandemic. This is because, considering the chronic nature of NCDs, they affect the quality of life, and result in premature deaths [8]. The RHNP comprised of three modules (water and nutrition, mother and child, and healthy lifestyles). The healthy lifestyle module consists of a healthy diet, exercise, drinking water, rest, and hygiene. The RHNP thus sought to increase the intake of fruits and vegetables, regular physical activity, encourage sufficient rest, and drinking adequate water among others to prevent the obesity category of the BMI status among the population and thereby reduce the risk to NCDs [9]. Health workers specifically community health nurses have been identified as important in the response to NCDs [10]. Thus, the choice of public health nursing (PHN) students also with the main responsibility of ensuring health promotion among the population (PHNs and community health nurses have core responsibility of health promotion. Difference is in qualification: PHNs are university degree holders whilst community health nurses have diploma from a non-university health institution) to understand barriers and motivators to improve these lifestyles among nursing trainees. Since they are young and also expected to cater for the population's health needs such as NCD reduction.

Findings from some contexts generally among young people, for example, Kulavic *et al* [11] identified self-motivation as a facilitator to physical activity among college students in the US. Among Iranian adolescents, issues of priority with studying, low self-esteem, lack of family and cultural support are some barriers to healthy eating [12]. Findings from young Australian males identified motivators to healthy diet and physical activity as physical appearance, physical health whilst cost and access, busy lifestyles, family upbringing, and peer influence were identified as barriers to physical activity and healthy eating [13]. A study by Seguin *et al* [14] in the US, identified lack of time and competing priorities, social norms, and stigma as perceived barriers to physical activity and healthy eating whiles social support, and ability to grow and produce food as facilitators respectively to physical activity and healthy eating among adults. Deliens *et al* [15] among students in a European university, identified some factors such as individual factors (e.g. taste preferences, self-discipline, time and convenience), their social

networks (e.g. (lack of) parental control, friends and peers), physical environment (e.g. availability and accessibility, appeal and prices of food products), macro environment (e.g. media and advertising) and university characteristics (e.g. residency, student societies, university lifestyle, and exams) to influence dietary behaviours.

In their study among young Ghanaians issues of maintaining chastity, safety, ill health, type of upbringing, and laziness were some barriers identified to physical activity [5]. Also, Tuakli-Wosornu *et al* [16] reported on physical activity motivators such as weight loss, health concerns, and increased energy. Associated barriers were, can't find the time, work and family obligations, and don't have a facility.

Despite the growing body of literature on perceived barriers and motivators/facilitators to lifestyle behaviours such as healthy dietary and physical activity in certain contexts among young people in the university, generally, to the best of knowledge, there is a paucity of information on young nurse trainees on the whole and specifically in Ghana. It is essential more health is obtained from individuals who are to provide healthcare. This study sought to use a qualitative approach to explore barriers and motivators to healthy dietary (fruit, vegetable intake, oil, salt use, eating time) and regular physical activity behaviours among nurse trainees (for this study, the concept of nurse trainees refers to pre-service individuals currently receiving formal training in a health training university specifically in public health nursing). Specifically, the research question to be addressed is, what barriers and motivators do nurse trainees experience in the practice of healthy dietary and physical activity behaviours?

## Materials and methods

### Study design

**Theoretical framework.**   The study sought to understand the enablers and barriers to healthy dietary and physical activity behaviours among young (18–25 years) pre-service nurse trainees. A qualitative approach using in-depth interviews (IDIs) (semi-structured interview guides) is employed to explore practice, barriers and motivators to healthy dietary and physical activity behaviours. This enabled the researchers to obtain in-depth information from the nurse trainees (specifically from trainees in the public health nursing track) regarding their experiences with dietary and physical activity in school.

**Participant selection.**   Prior to selection into the study, participants (trainees), were pre-informed of the study during lectures with a visit by two members of the research team. The class was informed of the study and the procedures it entailed. To avoid the disruption of lectures, the team was given a day in the following week when lectures have been cancelled for participants due to a school activity and we decided to utilize the said day. The trainees interested were informed to stay on after the school event to undergo the screening process. Participants (students) who were nursing trainees in the third year of a degree nursing (PHN) programme were given an equal chance to participate in the study. At the first stage, participants were screened and grouped based on their body mass index (BMI) status using the WHO classification. That is underweight ($<18.5$ kg/m$^2$), normal (18.5–24.9 kg/m$^2$), overweight (25.0–29.9 kg/m$^2$) and obese ($\geq30.0$ kg/m$^2$) groups. To do this, the anthropometric measures (weight in kg/ height in m$^2$) of participants were taken to determine which BMI status he or she belongs to. A bathroom scale was used to take weight measurements in kg and the height measurements were carried out initially in centimeters (cm) and converted into meters (m) with calibrations on a plain white wall. The team ensured all participants removed footwears and any other item that may influence the scale measurements.

Depending on the number within each group, participants were randomly selected at stage two given equal representation to males and females. However, using the BMI, there were five

underweights, two over weights, one obese and the rest were normal weight. Because there were more normal weights than needed, a randomization procedure that involved coin-flipping was applied to those with the normal BMI. A side of the coin specifically heads imply selection into the study thus, explaining how the eight normal weights were interviewed. Due to the fewer participants with the underweight, overweight, and obesity all were selected. Thus, a total of 16 IDIs were conducted. However, the study at the initial onset, decided to recruit as many participants for each BMI category as possible and only stop when data attains saturation.

**Setting.** The present study was carried out at the University of Health and Allied Sciences (UHAS). The institution admits students from all over the country. It is located in Ho, the capital of the Volta Region, and currently the sole public institution for training all cadres of health professionals.

Established by an Act of Parliament (Act 828 in December, 2011), the university started operations in September, 2012. Data collection for this study was conducted at the School of Nursing and Midwifery where the department of Public Health Nursing is located. Within the SoNAM, students within the public health nursing department were sampled since by training they are expected to work within communities to ensure health promotion activities including healthy lifestyles in NCD reduction compared to their counterparts in general nursing. This makes them more suitable for the study selection to understand issues about healthy dietary and physical activity (NCD risk factors) among them.

The SoNAM is hosted on the premises of the Ho Teaching Hospital (HTH) popularly known as Trafalgar after the construction company which built it. There are student residences on site for medical students whilst the SoNaM students stay in residences outside the campus. Majority are within walking distances to the campus. Whilst some lectures take place within the SoNAM, others are conducted about 25 minutes from the hospital campus to the location where the central administration is sited (permanent campus).

The participants are generally very mobile. Specifically, they are usually moving in or out of lecture halls, walking to campus or joining shuttles for lectures, and spending time together.

The interviews were conducted in September 2018 on the premises of the school of nursing and midwifery using a suitably offered space and they were arranged at the convenience of students in order not to disrupt their lectures.

**Data collection.** A two-day training was organised for four moderators. This was followed by pre-testing the interview guide provided by the researcher. Which was structured into five parts with opening questions, perceptions on BMI, assessment of lifestyle behaviours, facilitators to healthy lifestyles and barriers to healthy lifestyle behaviours.

The interviews were conducted in the English language, and audio recorded. Prior to the in-depth interviews (IDIs) a quantitative instrument to collect information on background factors was administered.

Participants were asked questions on such background variables [see screening questionnaire as a supplementary]: age (continuous variable), sex (male or female), residence (on-campus or off-campus), marital status (married or single), employment status (employed or student), religion (was open ended), height (captured in cm and converted to m) and weight (in kg).

Information on perceptions of BMI, self-assessment of lifestyle behaviours, perceived facilitators to the practice of lifestyle behaviours and perceived barriers to the practice of lifestyle behaviours were obtained from the participants [see IDI guide as a supplementary]. The lifestyles behaviours considered were dietary (fruit and vegetable intake, salt and oil use, and eating time), physical activity, hours of rest, water intake, smoking, and alcohol use. Out of the total questions asked only the responses to three are included in this paper on dietary and

physical activity behaviours: 1. Participants' self-assessment of dietary and physical activity behaviours, 2. Perceived facilitators to healthy diet and regular physical activity, 3. Perceived barriers to healthy diet and physical activity behaviours.

The interviews were digitally recorded and lasted for 50–60 minutes. The number of interviews was stopped based on saturation for the normal BMI group whilst for the other BMI categories the participants interviewed were those available. Interviews were scheduled on the morning following the measurements of the anthropometric measures prior to their first lecture of the day. Participants who could not be interviewed earlier, were scheduled in-between lectures. All participants were refreshed immediately after their interview session.

**Ethical considerations.** Ethical approval was given by the Research Ethics Committee (REC) (UHAS-REC A.6 [3] 17–18) of the University of Health and Allied Sciences. Administrative permission was then obtained from the Dean of SoNAM and written consent was obtained from all participants.

**Trustworthiness.** Trustworthiness in qualitative research involves credibility, transferability, dependability, and confirmability [17]. Credibility relates to the aspect of truth-value and is measured with strategies including the use of participants' words. With this, findings were supported with verbatim quotes from the trainees (participants). As such the individual transcripts were reviewed for similarities across participants and were grouped under the same themes.

Transferability is concerned with a thick description. This entails a detailed account is given on the context in which the study was conducted in addition to the description of the experiences and behaviours of study participants. This ensures that the experiences and behaviours of participants become meaningful to an outsider [17]. In the present study, transferability was ensured by describing the study setting, providing the sample size and sampling procedure used, and describing the socio-demographics of the study participants. Transferability was also ensured in the present study by providing the thematic table.

The focus of dependability and confirmability is on the audit trail [17]. The audit trail is about transparency in the description of the research processes from the beginning of a study to the development and reporting of the results. In the present study, the audit trail was ensured by documenting the entire research process from the background to the conclusion section. Regarding analyses, the thematic table has been provided. The interpretations of the data were also derived from the data collected and not based on the researcher's own preferences and viewpoints, which was made clear by providing a sub-section on reflexivity.

## Research team and reflexivity

**Personal characteristics.** Male or female in-depth interviews (IDIs) were conducted by any available interviewer irrespective of sex. The interviewers (three) were pursuing a master's degree and one already had attained a master's degree. These are individuals who were with the unit the researcher works. The researcher was at the time a Research Fellow. Prior to her Ph.D. and leading to the current position, the researcher had been a full-time research assistant.

**Relationship with participants.** The researcher who is a research fellow at the time also lectures and had lectured at SoNAM before. However, at the time of the study, the researcher had not lectured any group in SoNAM and thus, does not know participants and participants also do not know her. Reasons for conducting the research were made known to participants before engagement on the study by the interviewers (research team) who during training for data collection were introduced to the objectives/aims of the study. Participants were also informed the researcher is a faculty member of the institution.

## Analysis and findings

**Data analysis.** Interviews were transcribed verbatim by the interviewers. The reporting complied with the Consolidated criteria for reporting qualitative studies (COREQ).

The analysis was structured along the socio-ecological model [see Fig 1] to emphasise the importance of broader influences on dietary and physical activity behaviours of nurse trainees. Implying that, identifying motivators and barriers with a socio-ecological perspective may result in an improved means of enhancing and addressing barriers more specifically to improve poor dietary or physical inactivity behaviours. Since the model proposes that the manifestation of a behaviour is a result of several factors.

Initially formulated by Bronfenbrenner Urie [18], ecological models have included several factors with varying levels of influence on behaviour (intrapersonal/individual, interpersonal, organisational, community, and public policy) and considered the interaction of behaviours across these varying levels of influence. The present study, adapted the socio-ecological framework formulated by Story *et al* [19] with components on individual (intrapersonal), social

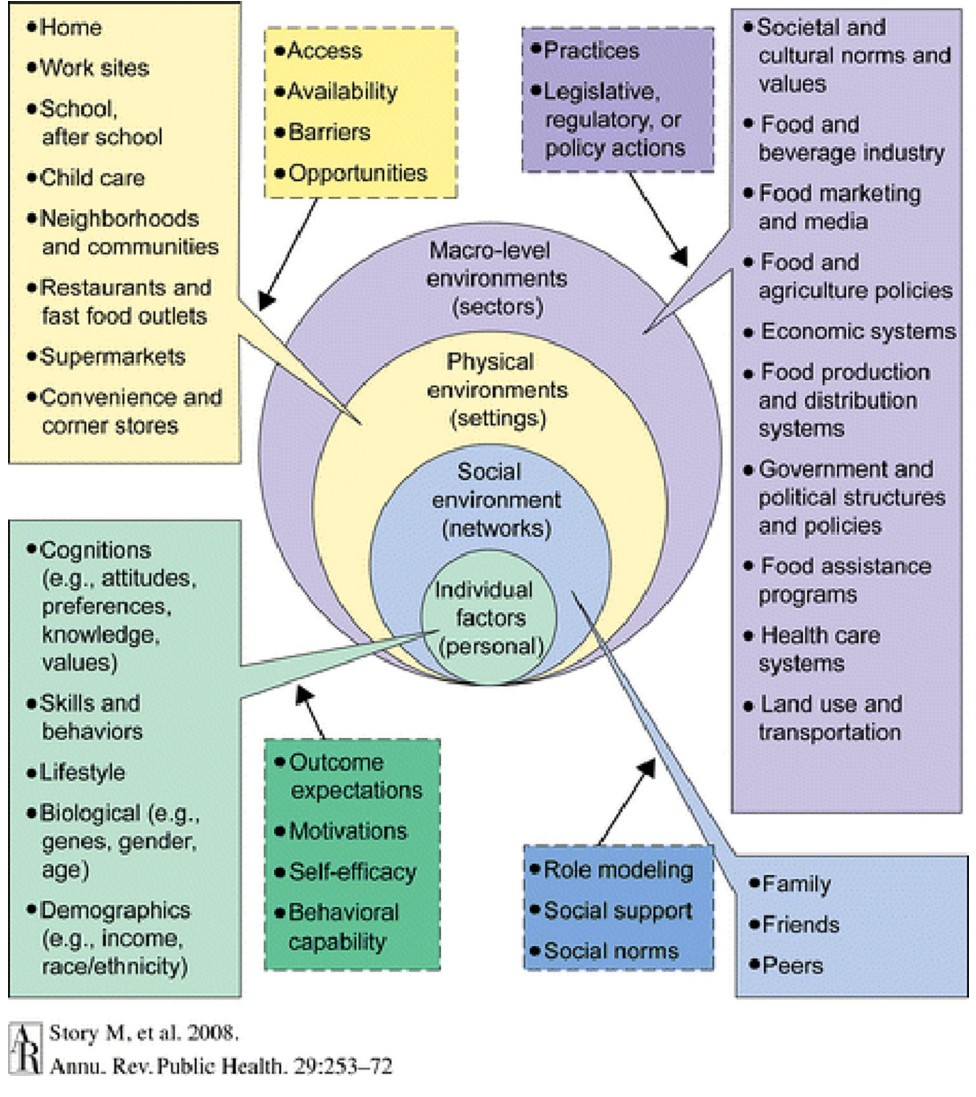

**Fig 1. Ecological framework from Story et al, 2008.**

(interpersonal) environmental, physical environmental and macro levels, to understand factors influencing eating and physical activity behaviours.

The reflexive thematic analysis was used [20]. Coding was carried out by the researcher and the Textual data was managed manually. The transcripts were read continuously for familiarisation. Codes were identified from the responses of the participants. The codes were then organized and arranged based on their shared patterns to form sub-themes (i.e. individual, social environment, physical environment, and university-related characteristics using a socio-ecological model) and then further grouped under the main themes. The main themes were identified in advance using the guide.

**Reporting.**   Quotations from participants were used to illustrate findings and the main themes were presented in the findings.

## Results

Presented are the background characteristics of the participants and results based on themes from the analysis.

### Background characteristics of participants

The sample (N = 16) distribution as shown in Table 1 consisted of 10 male and 6 female students with a higher proportion within the age range 21–25 years and a higher percentage with a normal BMI (50%).

**Table 1.  Socio-demographic characteristics of participants.**

| Background Characteristics | No. | Percentage (%) |
|---|---|---|
| **Age** | | |
| 18–20 | 6 | 37.5 |
| 21–25 | 10 | 62.5 |
| **Sex** | | |
| MALE | 10 | 62.5 |
| FEMALE | 6 | 37.5 |
| **Residence** | | |
| OFF-CAMPUS | 16 | 100 |
| **Marital Status** | | |
| SINGLE | 16 | 100 |
| **Employment Status** | | |
| STUDENT | 16 | 100 |
| **Religion** | | |
| CHRISTIAN | 16 | 100 |
| **BMI** | | |
| UNDERWEIGHT | 5 | 31.3 |
| NORMAL | 8 | 50.0 |
| OVERWEIGHT | 2 | 12.5 |
| OBESITY | 1 | 6.3 |
| **Total** | | |
| **No** | 16 | |
| **%** | | 100 |

*Source*: Fieldwork, 2018

## Participants' self-assessment of dietary and physical behaviours

**Assessment of dietary behaviour.**   Participants were asked their opinions with regards to their dietary behaviour (fruit, vegetable, salt, and oil intake and eating time) and physical activity. These questions were for participants to reflect upon and assess the practice of these select lifestyle behaviours.

Regarding dietary behaviour, while for some, it is not good and needed improvement others seem to be practicing healthy dietary behaviour.

*"vegetable intake I can say it is not all that good but as we in school we can have financial challenges and I can't afford a lot but any time I had some I try to buy some and eat"* (IDI, Male, normal BMI)

*"vegetables too the same unless we cook it in the house. . . . when I'm home I prepare it as a food in the house but when I come back to school I don't do it on my own. Like I like eehhh snack all these junk foods yeah I enjoy them most that is why"* (IDI, Female, underweight BMI)

*"I really don't take in salt"* (IDI, Female, normal BMI)

*"that one dieer I take oil, is much because, once I buy food outside dieer is much"* (IDI, Female, underweight BMI)

*"the oil intake too yeah I can see I take in much of oil because most of the foods everything is oil oil oil mostly in school preparing rice and stew, rice with oil, stew with oil, for egg you have to use oil, fish oil, I think it is too much "* (IDI, Female, normal BMI)

**Assessment of physical activity behaviour.**   In terms of physical activity, participants largely indicate a physically inactive behaviour as illustrated in the following quotes:

*"I don't indulge in any. Well because of my body size, I believe I'm fit so"* (IDI, Female, underweight BMI)

*"well let's say I don't have enough time to do that, because of lectures and other activities my time is really limited for me to do exercise."* (IDI, Male, Normal BMI)

*"I don't do anything"* (IDI, Female, underweight BMI)

However, although students walk some do not consider it a form of physical activity enough whilst others do. For instance:

*"I don't do much activity like physical activity and exercises and all. I only walk that's the exercise that I do"* (IDI, Male, underweight BMI)

*"As for physical activities it is okay for me because every day any lecture day I walk to the lecture for about let me say 10km"* (IDI, male, Normal BMI)

## Thematic results

Many factors influence dietary and physical activity thus the choice of the ecological model for analysis. According to the principles of the socio-ecological framework, a framework of factors (facilitators and barriers) was developed. It consists of individual, social environment, physical environment, and university characteristics. There are four main themes identified with a

**Table 2.  Themes.**

| Main themes | Sub-themes |
| --- | --- |
| Motivators for healthy dietary behaviour | |
| | Individual (Intrapersonal) Factors |
| | • Self-discipline |
| | • Dietary knowledge |
| | Social Environment |
| | • Social support |
| | Physical Environment |
| | • Geographical access/availability |
| Motivators for regular physical activity | |
| | Individual (Intrapersonal) Factors |
| | • Body image |
| | Social Environment |
| | • Social support |
| | University Characteristics |
| | • Existence of student societies |
| Barriers to healthy dietary behaviour | |
| | Social Environment |
| | • Upbringing |
| | • Preferences |
| | Physical Environment |
| | • Accessibility |
| | • Food safety |
| | University Characteristics |
| | • Studies/lectures |
| Barriers to regular physical activity | |
| | Individual (Intrapersonal) Factors |
| | • Busy lifestyle |
| | • Inadequate food satisfaction |
| | Social Environment |
| | • Upbringing |
| | • Social support |
| | University Characteristics |
| | • Studies/ Academic activities |
| | • Student societies |

varied number of sub-themes as shown in Table 2. To illustrate the sub-categories, the most appropriate quotes were chosen (see S1 File for additional quotes).

## Motivators for healthy dietary (fruit, and vegetable intake, salt, oil use and eating time) behaviour

**Individual (intrapersonal).**  *Self-discipline.* From the interviews, students indicate self-discipline to be related to healthy dietary behaviour. This is because they understand what healthy dietary behaviour is and its benefits. Thus, participants were of the opinion that there are no external facilitators needed to improve upon dietary behaviours of fruit and vegetable intake, oil and salt use and eating time. This is illustrated respectively in the following quotes:

*Fruit intake*

*"nothing really, I just have to make it like make it a point to be doing it [eating fruits]. There is nothing really like there is nothing actually"* (IDI, Female, underweight BMI)

*Oil use*

*no nothing can prevent me so far as I am the one to regulate it because even if I will prepare food I will be sure that the oil to use is low. Nothing will prevent me"* (IDI, Male, normal BMI)

**Dietary knowledge.** There is a need for knowledge on why certain foods should be included or excluded in a diet. Providing knowledge on the benefits of eating fruits is essential to facilitate its intake in addition to healthy ways of preparing meals. This knowledge will ensure the inclusion of fruits and vegetables in diet while limiting oil and salt use, and eating on time:

*"If I see and know the essence of taking in fruits. Because some fruits are not common here and they are rare to come by so if you tell me that ohh taking fruits will give me a good skin like because I know what it will give me I will take it"* (IDI, Female, normal BMI)

Also, new ways of doing things as indicated specifically in terms of new healthy methods of cooking may propel an individual to a healthier lifestyle as illustrated in the quote:

*"I think that the oil intake if there is a substitute for the oil may be using more alternative means instead of using oil all the time. . .oil but if there is some substituent to it not to always take oil may be frying of the egg or maybe there is a measure it that if you are frying egg one egg you have to use this quantity of oil something like that so that it should be measurable. Because you just pour it into the frying pan and by the time you realise [laughter] you just go"* (IDI, Female, normal BMI)

**Social environment.** *Social support*. Considering the Ghanaian setting with its wide network of family and friends, individuals generally look out for available support when engaging in activities. This support translates into approval or disapproval. This is important in promoting healthy dietary behaviour:

*"Actually, my family doesn't like too much salt so that has been preventing me that has motivated me not to eat too much salt"* (IDI, Female, underweight BMI)

**Physical environment.** *Geographical access/availability*. The location participants find themselves is a means to practice a healthy lifestyle. As this impacts the availability and geographical access to fruits and/or vegetables. In other words, this may improve or deny access to available fruits/vegetables thus impacting dietary behaviour. This is illustrated in the following quotes:

*"okay if I'm not seeing them around and if I don't feel like the urge is not there for me to eat I will never eat. but if I don't see them if I see eee like banana around, I like banana a lot so if I see banana right now it will motivate me to buy that's all. If I see the fruit around I will eat it but if I don't see them* (IDI, Female, underweight BMI)

## Motivators for engaging in physical activity

**Individual (intrapersonal).** *Body image*. The desire to acquire a particular body size or the perception that physical activity is for attaining a particular body size can influence one's physical activity level as illustrated in the following quote:

> "*I think it has all do to with one's self. For someone like me I wouldn't say physical activity is much needed for me because I want to gain more weight. I don't want to lose. So, I think it has to do with one's own perception of uh. . . physical activity. If the person sees he has gained too much of weight then he needs to engage in more physical activity*." (IDI, Male, underweight BMI)

**Social environment.** *Social support*. The presence of family/friends in the physical activity journey is perceived as essential. The students have revealed that support from others in the form of encouragement, or just being there can influence physical activity behaviour. This is illustrated in the following quotes:

> "*I always want encouragement and company when I have these two and I have someone to follow me down this road every morning I will do that and even if there is someone to motivate I will do, but now I am alone.*" (IDI, Female, Obese BMI)

> "*My friends, when I see them doing it, I am pushed to do it*" (IDI, Male, Normal BMI)

**University characteristics.** *Existence of student societies*. The existence and belonging to organized groups/societies on university campuses form part of the university life. Functional organised groups or societies on campus for physical activities can be a facilitating factor to improve regular physical activity among students since it brings people together promoting regular physical activity. Also, it tends to allay the fear of exercising alone and offers support:

> "*I really like exercising so when I get home, I usually exercise but when I come to school no group or no association really enforcing that on campus. So just come maybe unless this denomination they are having a programme then they include it into their activities. But specifically let's say every Saturday morning we are going on jogging no we don't have that type of activity on campus. . . . if it's also enforced on campus that every Saturday morning we jog from here to here you come back and have some acrobatics I think it will really help*" (IDI, Female, normal BMI)

## Barriers to healthy dietary behaviour

**Social environment.** *Upbringing*. The Ghanaian socialization process in terms of meals is not centered around fruits and vegetables but carbohydrates. However, there is a lot more inclination towards vegetable intake as the sauces accompanying the carbohydrate meals have vegetables in the recipes. In terms of consuming fruits, a participant indicates a dislike as they are seen as not fulfilling:

> "*As for the vegetables, I take but the fruits I don't know I don't like it that much*" (IDI, Male, underweight BMI)

*Preferences*. Preferences for particular fruits or vegetables determine a dietary behaviour inclusive of fruits/vegetables. This tends to restrict people to only certain fruits/vegetables and exclude others when the preferred is not available:

"*okay the vegetables especially nkotomire my throat will be itching me. Apart from the carrot, cabbage, that one is okay with me. That is why it doesn't motivate me to eat nkontomire stew*" (IDI, male, underweight BMI)

Some participants indicate nothing will prevent them from healthy dietary behaviour as the lack or absence of fruits and or vegetables from their diet is due to dislike and nothing else:

"*nothing I just don't like it that's all*" (IDI, Male, underweight BMI)

### Physical environment

**Accessibility.**   The geographical and financial access to fruits and vegetables prevents its adequate consumption by students. This is illustrated in the following:

"*. . .where to get the fruits cos if you close from lectures very late and then like 6.30pm you can't get it around anywhere you have go to the market so it been near to you and then financial constraint.*" (IDI, Female, normal BMI)

**Food safety.**   Safety of food particularly fruit or vegetable is a source of concern. This is because, individuals do not know or are not clear on how it was cultivated, and or prepared. Since its appearance after preparation sometimes turn people off. People naturally are attracted to well-presented foods:

"*how it appears after cooking will prevent me from eating. If it is not attractive, I'll not take or even if it is not prepared well I won't take it*" (IDI, Female, Normal BMI)

"*Because recently there's been a news around that growing those eerrr fruits and vegetables they've been they don't wait for eerrr I don't know how to put it. Like they've been using fertilizers right? And too much fertilizer will pose health risk to we the humans and I think in our communities like if they grow the crop it should take 30 days for the vegetables to be ripe aahaa instead of them waiting for that they go in for the fertilisers to decrease the time range the normal time range the vegetables need to get ripe. . . . So, to me I don't think it will be good for me. I like the way I am*" (IDI, Male, normal BMI)

**University characteristics.**   *Studies/lectures*. The core reason for being in school i.e. studies has been identified as an obstacle to healthy dietary behaviour as students seem not to have time due to studies.

In terms of regular eating times, studies emerged as an obstacle as students leave early for lectures, stay on for group discussions or attend re-scheduled lectures among others:

"*my studies, coming for lectures that's all*" (IDI, Male, underweight BMI)

"*the only thing that may prevent me from taking less oil is the time to cook the dish that can do without oil. If there is time maybe I can but if there is no time I have to go buy food. . .*" (IDI, Female, Normal BMI)

### Barriers to regular physical activity

**Individual (intrapersonal).**   *Busy lifestyle*. Insufficient time is indicated as a barrier to regular physical activity as individuals indicate they are too occupied with academic activities to take up physical activity. This is illustrated in the quote below:

"*okay I don't have the time to do it that's all. . . I come for lectures, during weekends I do chores, I study so I don't get much time*" (IDI, Male, Underweight BMI)

*Inadequate food satisfaction.* Physical activity is perceived to be taken by persons who are adequately fed. A participant perceives the need to be fully satisfied before attempting to engage in physical activity. This could imply physical activity is a reserve for those who have enough to eat.

". . .*if you are like I have already said if eermm there is no food you will not get any energy that you are hungry you can't exercise otherwise you run at hypoglycemia*" (IDI, Male, Normal BMI)

**University characteristics.**   *Studies/academic activities.* Academic-related activities have been reported to obstruct regular physical activity. Issues such as tiredness emanating from tight school schedules or sometimes inadequate time due to activities associated with studies. This is illustrated in the following:

"*Every day we go for lectures from Mondays to Fridays and the lectures start from 7.30 am and we have to go to the bus station 6.30am for the bus to come and pick you for lectures and Saturdays too Saturday is the ideal time for us to exercise but I don't exercise. I feel lazy.*" (IDI, Female, Underweight BMI)

*Student societies.* Various student associations/groups on campus usually organize outdoor activities such as trekking, mountain climbing, sporting activities to promote unity among members. This encourages physical activity among students who otherwise may not have initiated or sustained this lifestyle:

"*Yes when I am on campus I don't exercise, the only time I exercise is when maybe a religious organization or class want to play football match, that's the only time I will exercise. And it comes once in a while. Last semester I tried exercising but I couldn't devote my time, just once, twice and I stopped. I brought my training kits this semester for jogging but am still not doing it.*" (IDI, Male, Underweight BMI)

**Social environment.**   *Upbringing.* Physical inactivity has been attributed to the socialization process. A part of the socialization process is for individuals to stay put when they do not have much to carry out. Since people who move around are mostly regarded as recalcitrant. As a result, individuals are unable to internalize a physically active behaviour:

"*I'm very lazy so maybe we planned to go jogging around 4.30am as for my sleep dieer I don't want to eeheee if it comes to my sleep waking up for the jogging I can't and I think it's okay*" (IDI, Male, Normal BMI)

"*When I don't have anything doing I stay indoors so that is the only thing that will prevent me from walking. But apart from that when I have lectures I walk and when I want to buy something from the town I walk to the market to buy it. But when I am at home and don't feel like doing anything that means I am indoor.*" (IDI, Male, Overweight BMI)

*Social support.* Initiating and sustaining physical activity behaviour requires encouragement from family, friends, and significant others. The absence of this support system may present physical activity as not important to individuals who desire to participate:

"*No, there is no barrier apart from company and tiredness.*" (IDI, Female, Obese)

## Discussion

The purpose of this qualitative study is to examine perceived enablers and barriers to healthy dietary and physical activity behaviours among pre-service nurse trainees. Factors identified to influence dietary behaviour and physical activity were on levels of: individual, social environment, physical environment and some factors of higher institutions of learning (University characteristics). The factors facilitating healthy dietary behaviour included: self-discipline, dietary knowledge, social support, and geographical access/availability. Facilitators of regular physical activity includes: body image, social support and student societies. In relation to perceived barriers affecting healthy dietary behaviour, participants mentioned: upbringing, preference, accessibility, safety, studies/lectures whilst that of physical activity includes: busy lifestyle, inadequate food satisfaction, studies/academic activities, student societies, upbringing and social support.

### Self-assessment of dietary and physical activity behaviours

Findings of low fruit and vegetable intake and high oil consumption but low salt usage emanated from participants' self-assessment of their dietary behaviour. This supports findings from Amoateng *et al* [3] on Ghanaian youth, which indicate poor intake of fruits and vegetables. The poor intake of fruits and vegetables among students may be attributable to the general perception that fruits and vegetables are not filling as compared to carbohydrate-based foods that dominate the Ghanaian diet. Also, being in school, students are usually financially dependent and will not perceive it prudent to spend money on what to them is not filling. For the high intake of oil, students tend to spend less time cooking and their fast recipe usually is boiling a mixture of oil and rice together. Sometimes insufficient time propels them to buy food from vendors whose gravy (tomato-based accompaniment to the carbohydrate component) is usually dominant with oil.

In general, students perceive themselves physically inactive as they do not undertake regular physical activity. Although some walk to campus (Southern campus) they usually do not consider this as physically active since such walks are done leisurely. Also, because walking has become part of daily routine, individuals no longer consider it as physical activity.

### Perceived enablers to healthy dietary behaviour

Perceived enablers to healthy dietary behaviour identified in the present study are consistent with findings of Deliens *et al* [15] where university students also reported self-discipline, dietary knowledge, social support, and accessibility as facilitators to healthy dietary behaviour. As students in a health institution, it is clear information on healthy dietary behaviour when provided will change lifestyles positively as well as an individual's self-discipline. Evidence from HIV/AIDS literature showed knowledge does not impact behavioural change as such the students mentioned social support. Social support as identified from a previous study is important for maintaining a healthy dietary behaviour [14]. These individuals (social support) can serve as reminders to participants to be conscious of a better dietary lifestyle in terms of fruit, vegetable, salt, oil use, and eating time. As students share place of abode (with colleagues or

family members in town), their dietary behaviours are sometimes influenced by these significant others. Being a school setting, individuals indicate geographical and financial access to the right food environment will aid a better dietary behaviour. The town (Ho) practices an open market system with market day rotating on various days (every four days) it is not assured individuals will have access to fresh foodstuffs they may need on regular basis outside the market days. Therefore, access to foodstuffs may enhance or promote a healthy lifestyle.

## Perceived enablers to physical activity

The perceived enablers to physical activity among participants were body image, social support, and student societies. The finding on body image is consistent with a study by Doegah *et al* [5], Ashton *et al* [13], Kulavic *et al* [11] and Ashton *et al* [21] among young Ghanaians, Australians, Americans and Australians respectively where physical appearance, weight management and body image emerged as a motivating factor to engage in physical activity. This appears to becoming a norm in the Ghanaian society such that underweight and normal-weight individuals seem to socially self-exempt themselves from physical activity. Thus, the manifest reason for physical activity is yet to be established in the context.

Participants perceived the presence of significant others (social support) to act as an enabling factor for physically activity. This corroborates a finding by Kulavic *et al* [11]. Most people tend to walk or jog and usually undertake this activity at dawn or evening. Thus, significant others can be for companionship and safety as others prey on students when they are alone. This is confirmed in a study by Doegah *et al* [5] where safety emerged as a barrier to physical activity among young Ghanaians. Thus, to undertake physical activity alone can be clouded with issues of security and sometimes the intention to get it started may never materialize. Therefore, some students were of the opinion that the presence of organized bodies (societies) will help initiate physical activities and keep it ongoing among them.

## Perceived barriers to healthy dietary behaviour

The perceived barriers influencing dietary behaviour in the present study support other research findings such as upbringing, preference, accessibility [13], safety/appearance, and studies/lectures [15]; Musaiger *et al* [22].

In terms of upbringing, some participants indicate they consume vegetables than fruits. Most Ghanaian dishes that serve as an accompaniment to the carbohydrate meal are usually prepared from vegetables and may explain the inclination towards vegetables than fruits. Regarding accessibility, students are financially dependent on parents/guardians for their upkeep. Thus, it is expected they utilize the least they have been offered. This corroborates findings by Sogari *et al* [23]; Amore *et al* [24] among college students. In addition to financial access, geographical access to fruits, and vegetables impacts healthy dietary behaviour. Because, students usually have to visit the market to get fruits/vegetables fresh and cheap as sellers bring them from the surrounding villages compared to those on campus or around their places of abode.

Food safety in terms of fruits/vegetables is crucial since some people perceive the already prepared ones not safe for instance not well washed and sometimes the appearance of vegetables after cooking turns off people as it appeared no longer attractive. Closely associated with safety is the means of cultivating some fruits and vegetables. There are unverified concerns about bad farming practices within the Ghanaian context. Some of these unverified concerns were the use of waste water for watering, fertilizers or esters for early ripening (anecdotal source). These unsubstantiated claims obstruct the intake of fruits and or vegetables. As individuals fear these practices though alleged affect the produce and may be a source of ill health.

Also, activities usually associated with education (i.e. studies) tend to prevent participants from engaging in healthy dietary behaviour [15]. In order to have enough time for academic related activities (study/lectures) students are also unable to eat well or on time or cook healthily with less oil or salt.

## Perceived barriers to regular physical activity

Factors such as busy lifestyle, inadequate food satisfaction, studies/academic activities, student societies, upbringing and social support have been mentioned by participants as obstacles to regular physical activity amongst them.

The finding on busy lifestyle collaborates that of Doegah *et al* [5]; Seguin *et al* [14]; Ashton *et al* [13]; Musaiger *et al* [22]; Ashton *et al* [21]; Anjali *et al* [25] where busy lifestyles/lack of time/ time constraint were obstructing physical activity. These are students who indicated much study workload with minimal free time at hand and are also constraint with time since they are regularly conveyed from one campus to another for lectures/interim assessments (observational). All these time expectations may obstruct them from a physically active lifestyle. An issue of inadequate food satisfaction also emerged. Since strength is required for physical activity individuals who think they do not get enough to eat have forbidden themselves from physical activity. Closely associated with this, is the financial constraints that hamper participants' desire to eat well. As such some engage in eating formulae such as 'one, zero, one' or 'zero, one, zero' (where 'zero' refers to food abstinence and 'one' refers to eating). Consequently, the inability to have the required strength will result in physical inactivity. On the other hand, some individuals are starving themselves as a means to acquire a desired weight size and therefore such individuals may think they are already achieving their goal and have no need for physical activity.

University-related issues of studies/academic activities and absence of student societies have been attributed to physical inactivity. Because the core business for participants is to acquire a certificate at the end of their training, most maybe reluctant to engage in activities that seem irrelevant to the degree. Focus tends to be on studying for exams and thus promoting a sedentary lifestyle. This finding is similar to that of Wang *et al* [26]; Sogari *et al* [23]; Rajaraman et al [27]; Deliens *et al* [28]; who reported high sedentary lifestyle among university students due to studies. In addition to studies, the absence of student groups on campus that may encourage physical activity appears missing. This may be attributed to the singular attention on academic activities to the detriment of other activities.

From the present study, the socialization process which has to do with upbringing tends to obstruct a physically active lifestyle and this is consistent with a finding by Doegah *et al* [5]. The socialization process is so imbibed that pre-service nurse trainees, could not alter this behaviour. Also, the absence of social support in terms of physical activity has been mentioned as a preventive factor. This is similar to a finding by Ashton *et al* [21]; Musaiger [22]; Anjali *et al* [25]. Since physical activity is difficult to start and sustain (observation) the presence of others to ensure it is actually taken up is essential for a physically active lifestyle. However, within the university context where focus is on studies, students or parents may not be interested in how physically active colleague students or wards are in school.

**Limitations and strength.**   A limitation to the present study is that, these results cannot be generalized to the whole population of university students as the data was collected within only one school but has generally enlightened us on enablers and barriers perceived to influence healthy dietary and physical activity of pre-service nursing trainees. Also, the study's data were based on self-reports by the trainees, so it is subject to possible social desirability responses. However, findings compare with earlier studies in other contexts. Another

limitation relates to the trainees. The trainees who participated in the study knew what the study is about. Since, the anthropometric measures were taken a day before the actual interview, trainees who were possibly comfortable with theirs reported the following day for recruitment into the study which may be considered as selection bias. This may explain the high turnout rate of trainees with the normal BMI status. A strength of this study is that, data was collected from participants based on BMI status.

## Conclusion

This present study examined perceived motivators and barriers to dietary and physical activity among public health nursing trainees. Applying the socio-ecological model, motivators and barriers were identified with regards to the individual (Intrapersonal), social environment, physical environment and university related characteristics. The results thus, indicate influencers for healthy lifestyles specifically dietary and physical activity behaviours are of multi levels. Therefore, to promote facilitators and address barriers requires the consideration of all levels identified from the results. An intervention such as a program to support healthy lifestyles for nursing trainees is needed in consideration of all the levels of the socio-ecological model. A research question that readily comes to mind is, what other modifiable NCD risk behaviours are nurse trainees not practicing well? Future research should focus on profiling NCD risk factors in higher educational institutions in Ghana particularly with focus on health trainees considering they will work for our health one day and must engage in healthy lifestyles to promote such among the population.

## Supporting information

**S1 File. Supplementary quotes from participants' response.**
(DOCX)

**S2 File. Screening questionnaire.**
(DOCX)

**S3 File. IDI guide.**
(DOCX)

## Acknowledgments

We wish to thank all interviewers and interviewees for participating in this study. We also thank the editor and the reviewer whose inputs have contributed greatly to improve the quality of the manuscript.

## Author Contributions

**Conceptualization:** Phidelia Theresa Doegah.

**Formal analysis:** Phidelia Theresa Doegah.

**Funding acquisition:** Phidelia Theresa Doegah.

**Investigation:** Phidelia Theresa Doegah, Evelyn Acquah.

**Methodology:** Phidelia Theresa Doegah.

**Project administration:** Phidelia Theresa Doegah.

**Supervision:** Phidelia Theresa Doegah.

**Writing – original draft:** Phidelia Theresa Doegah.

**Writing – review & editing:** Phidelia Theresa Doegah.

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
