## [Decision Letter · Decision Letter 0]

31 Mar 2022

PONE-D-22-03540Promoting healthy lifestyles among nurse trainees: perceptions on enablers and barriers to dietary and physical activity behavioursPLOS ONE

Dear Dr. Doegah,

Thank you for submitting your manuscript to PLOS ONE. After careful consideration, we feel that it has merit but does not fully meet PLOS ONE’s publication criteria as it currently stands. Therefore, we invite you to submit a revised version of the manuscript that addresses the points raised during the review process.

We look forward to receiving your revised manuscript.

Kind regards,

Michael B. Steinborn, PhD

Academic Editor

PLOS ONE

Journal Requirements:

3. We noted in your submission details that a portion of your manuscript may have been presented or published elsewhere. 

(PLOS Global Public Health, Full submission, PGPH-D-21-00284)

Reviewers' comments:

Reviewer's Responses to Questions

**Comments to the Author**

1. Is the manuscript technically sound, and do the data support the conclusions?

Reviewer #1: Partly

2. Has the statistical analysis been performed appropriately and rigorously? 

Reviewer #1: N/A

3. Have the authors made all data underlying the findings in their manuscript fully available?

Reviewer #1: No

4. Is the manuscript presented in an intelligible fashion and written in standard English?

Reviewer #1: Yes

5. Review Comments to the Author

Reviewer #1: Thank you for the opportunity to review the manuscript ‘Promoting healthy lifestyles among nurse trainees: perceptions on enablers and barriers to dietary and physical activity behaviours‘.

This paper investigates in perceived motivators and barriers to dietary and physical activity among public health nursing trainees in Ghana. As a part of a qualitative approach, the authors used the socio-ecological model to identify motivators and barriers with regards to the individual, social environment, physical environment and university related characteristics, assessed in in-depth interviews.

Enablers to healthy dietary behaviour were self-discipline, dietary knowledge, social support and access/availability. Barriers to healthy dietary behaviour included upbringing, preference, accessibility, safety/appearance and studies/lectures. Enablers related to physical activity mentioned were body image, social support and the existence of student societies. Barriers mentioned in relation to physical activity include a busy lifestyle, inadequate feeding, studies/academic activity, student societies, upbringing and social support.

The study provides new insights in influencers for healthy lifestyles specifically dietary and physical activity behaviours and potentials to promote facilitators and address barriers on multi levels.

In general, the manuscript is written in a comprehensible way. However the research question and hypotheses should be emphasized more in the introduction and a great deal of attention should still be paid to the methods, results and discussion section. There is a lack of information on how and with which tools the evaluation of the data was carried out, which should definitely be specified. The presentation and discussion of the results and its concluding limitations and strengths still need improvement. Also there are some minor issues described in detail below.

Abstract

The abstract and title summarize the most important parts of the study in an appropriate way.

I would recommend to specify the design of the study and the method of in-depth interviews. Furthermore, in lines 52 to 53, it says ‘Enablers and barriers were grouped into levels […] based on the ecological model’. Here I would recommend to add Bronfenbrenner as the author of the ecological model.

Introduction

All in all, the introduction introduces the topic, gives an overview of previous research results and identifies gaps in the research literature. Nevertheless, it is not made clear why the study specifically refers to nursing trainees.

In lines 71 to 73 it says that ‘excess salt use, and physical inactivity have contributed respectively to 4.1 million, and 1.6 million deaths annually’. I think it would be important to add that this is the number of deaths worldwide and give the source of the information. Moreover, there is a space missing in line 72 ‘4.1million’.

In lines 94 to 97, the connection is made between the health needs of the population and study of nursing trainees. This connection could be elaborated a little further to emphasize the relevance of the sample.

I would also recommend to establish a stronger theoretical framework with the named findings from lines 99 to 119 more closely related to the research question and purpose of the study. Although the aim of the study is stated at the end of the introduction, no clear research question or hypothesis is given. This should be specified.

Materials and Methods

The methods section is very detailed with a lot of irrelevant information. Nevertheless, important information is missing. Especially the interview method could be described in more detail, e.g. Which questions were specifically asked in the interview? How exactly is the interview guide structured? What program was used to analyze the data?

In the study design section there is no information if there was a power analysis calculated prior to data collection to determine the necessary sample size. Further questions remain unanswered in the description of sample selection: How exactly did the participants learn about the study (e.g. in class, by mail)? Was participation in the study mandatory or voluntary? Was there any remuneration (e.g. voucher, money etc.)?

Especially the setting section with the detailed information about the university is very extensive and should be reduced to the most important information. I stumbled over the statement in the lines 168 to 171. Here one should justify more why these trainees fit better for the study than trainees in general nursing. I would also recommend to shorten the detailed description of the SoNAM and its localization to the most relevant information.

Single-digit numbers, like ‘4’ in line 192 (also line 339), should be written out, according to the current APA guidelines (APA Publication Manual, 7th ed.). When describing the variables collected in line 196, it is important to not just list them as examples, but to report everything that was collected.

In the section ‘Research team and reflexivity’ I would be interested to know if the interviewers were aware of the content of the study, i.e. if they were blind/non-blind (in the sense of involved in the study or not). In lines 203 to 204, it is stated that only the responses to three of the total questions of the in-depth interviews are included in the paper. It will become clearer for the reader if it is described in more detail which questions are explicitly meant. To make it easier for readers to understand the ecological model underlying the analysis, a graphical representation would help to visualize.

To sum up, I would recommend adding more important information and reduce irrelevant from the materials and methods section so that reader can better understand the methodological implementation.

Results

All in all, the results section is very extensive and should be more concise. That’s why I found it difficult to identify the main findings of the study and to differentiate between the relevant and irrelevant points of the section.

Care should be taken to report statistical values in an appropriate manner. In line 282 the sample size is given as ‘n = 16’. The current APA guidelines recommend ‘n = 16’ instead. Likewise, the table in lines 287 to 288 should be adjusted according to APA guidelines. One aspect I stumbled upon is the assessment of the behavioural variables (diet and physical activity) in the results section, which would be better housed in the methods section to make the collection of the data more understandable. The thematic results should be better structured and visualized, for example with the help of figures or tables. Moreover, I would recommend to reduce the examples of illustrating quotes of participants answers, embed them more into the text or to add them to the appendix, if necessary.

Discussion

The discussion of the key results of the study is very extensive and should be summarized more concisely and presented coherently. Due to the quantity of findings it could also become clearer to the reader if there would be subheadings.

I would also advise to be careful with statements like in lines 748 to 750, because they are not scientifically provable. In line 752 an exact reference is needed.

The limitations of the study are mentioned, but could be more extended by taking into account sources of potential bias or imprecision. What’s missing is a detailed discussion of the generalization of the study results as well as the strengths of the study. In lines 795 it would be more understandable if the authors explain why the results cannot be generalized. Is it due to the sample size and/or cultural and other differences between African and, e.g., European countries, etc? In addition, interviews may be subject to bias (e.g. socially desirable responses) because only the subjective experience is recorded.

Conclusion

In the abstract, there is mention of the need for a program to support trainees. However, this information is missing from the conclusion section of the manuscript. Also missing is information about new research questions and further studies that should be conducted building on these findings. Overall, the conclusion lacks the bottom line of what we can draw from the study results.

6. PLOS authors have the option to publish the peer review history of their article (what does this mean?). If published, this will include your full peer review and any attached files.

Reviewer #1: No

---

## [Author Response · Author response to Decision Letter 0]

15 May 2022

Editor’s Comments 

Response: We have reformatted our manuscript to meet PLoS ONE journal requirements. 

Response: As suggested on the funding information, we have provided the website for the Ghana Studies Association and provided a statement regarding the fact that, it is not a usual funding agency but provides such grants as research support to early researchers. Thus, there are no grant numbers.

3. We noted in your submission details that a portion of your manuscript may have been presented or published elsewhere. 

(PLOS Global Public Health, Full submission, PGPH-D-21-00284)

Response: What we indicated with regards to the PLOS Global Public Health was not peer-reviewed and formally published. It was an earlier version of the manuscript. Authors confirm this manuscript submitted to PLoS ONE has not been published or is under consideration elsewhere. 

a.) If there are ethical or legal restrictions on sharing a de-identified data set, please explain them in detail (e.g., data contain potentially sensitive information, data are owned by a third-party organization, etc.) and who has imposed them (e.g., an ethics committee). Please also provide contact information for a data access committee, ethics committee, or other institutional body to which data requests may be sent. 

(there are ethical issues with sharing the de-identified data set. 

Response: The data contains sensitive information that when obtained can be traced to participants. Data will be available upon reasonable request from the ethics committee as follows: Mr. Fidelis Anumu (Ethics committee administrator, University of Health and Allied Sciences, rec@uhas.edu.gh

Response to Review Comments

S/N Comments Responses

Abstract

1. I would recommend to specify the design of the study and the method of in-depth interviews. 

Response: Thank you for your comment. We have specified the study design and method of IDIs in the abstract. Please see lines 47 to 48. 

2. Furthermore, in lines 52 to 53, it says ‘Enablers and barriers were grouped into levels […] based on the ecological model’. Here I would recommend to add Bronfenbrenner as the author of the ecological model. 

Response: Thank you for your suggestion. We have duly attributed the ecological model to Bronfenbrenner in the abstract. Please see line 55.

Introduction 

1. In lines 71 to 73 it says that ‘excess salt use, and physical inactivity have contributed respectively to 4.1 million, and 1.6 million deaths annually’. I think it would be important to add that this is the number of deaths worldwide and give the source of the information. Moreover, there is a space missing in line 72 ‘4.1million’. Response: Thank you for this observation. The information in line 71 to 73 have been rewritten as a global statistic and referred. Please see lines 72 to 75.

2. In lines 94 to 97, the connection is made between the health needs of the population and study of nursing trainees. This connection could be elaborated a little further to emphasize the relevance of the sample. 

Response: Further justification has been provided to emphasize the sample relevance. Kindly see lines104 to 121.

3. I would also recommend to establish a stronger theoretical framework with the named findings from lines 99 to 119 more closely related to the research question and purpose of the study. 

Response: A stronger theoretical framework together with the research question has been stated on lines148 to 152.

4. Although the aim of the study is stated at the end of the introduction, no clear research question or hypothesis is given. This should be specified. 

Response: Thank you for this suggestion. The study’s research question is stated on lines 156 to 158. 

Materials and methods

1. The methods section is very detailed with a lot of irrelevant information. Nevertheless, important information is missing. Especially the interview method could be described in more detail, e.g. Which questions were specifically asked in the interview? How exactly is the interview guide structured? What program was used to analyze the data? 

Response: Thank you for this question. Please information on questions asked in the interview (please see lines 257 to 262) and how the interview guide was structured (please see lines 244 to 246) are provided in the section on data collection. 

Please, the data was analysed manually and stated under the section analysis. Please see line 342 to 343. 

2. In the study design section there is no information if there was a power analysis calculated prior to data collection to determine the necessary sample size. Response: Thanks for the comment. Please, there was a plan towards recruitment until data reaches saturation. Please refer to lines 196 to 199. 

3. Further questions remain unanswered in the description of sample selection: How exactly did the participants learn about the study (e.g. in class, by mail)? Was participation in the study mandatory or voluntary? Was there any remuneration (e.g. voucher, money etc.)? 

Response: Thank you for the question. Please, clarity has been provided on all these in lines 173 to 178.

4. Especially the setting section with the detailed information about the university is very extensive and should be reduced to the most important information. 

Response: Thank you for your suggestion. This section has been reduced. Please, see the section on setting. 

5. I stumbled over the statement in the lines 168 to 171. Here one should justify more why these trainees fit better for the study than trainees in general nursing. Response: Thank you for the suggestion. Additional justification has been provided. Please see lines 215 to 220.

5. I would also recommend to shorten the detailed description of the SoNAM and its localization to the most relevant information. 

Response: Thank you for the suggestion. The detailed description on SoNAM has been trimmed down to essential information. Kindly see the section on setting. 

6. Single-digit numbers, like ‘4’ in line 192 (also line 339), should be written out, according to the current APA guidelines (APA Publication Manual, 7th ed.). 

Response: Thank you for pointing this out. Single-digit numbers throughout the manuscript have been presented according to the APA Publication Manual, 7th ed. Please see lines 189 to 194, 243, 263, 310.

7. When describing the variables collected in line 196, it is important to not just list them as examples, but to report everything that was collected. 

Response: Thank you for the comment. Everything collected on these variables has been clearly stated. Please see lines 253 to 256.

8. In the section ‘Research team and reflexivity’ I would be interested to know if the interviewers were aware of the content of the study, i.e. if they were blind/non-blind (in the sense of involved in the study or not). 

Response: Thank you for the question. Actually, the interviewers were aware of the content of the study. A statement to this has been provided on lines 319 to 321. 

9. In lines 203 to 204, it is stated that only the responses to three of the total questions of the in-depth interviews are included in the paper. It will become clearer for the reader if it is described in more detail which questions are explicitly meant. Response: Thank you for this question. The questions have been clearly stated. Kindly see lines 262 to 266. 

10. To make it easier for readers to understand the ecological model underlying the analysis, a graphical representation would help to visualize. 

Response: Thank you for your comment. A figure has been provided. Kindly see page 12 (lines 349 to 350). 

Results

1. Care should be taken to report statistical values in an appropriate manner. In line 282 the sample size is given as ‘n = 16’. The current APA guidelines recommend ‘n = 16’ instead. 

Response: Thank you for pointing this out. The correction has been made according to the APA format. Please see line 361. 

2. Likewise, the table in lines 287 to 288 should be adjusted according to APA guidelines. 

Response: Thank you for the comment. The table has been restructured according to the APA format. Please see lines 366 to 368. 

3. One aspect I stumbled upon is the assessment of the behavioural variables (diet and physical activity) in the results section, which would be better housed in the methods section to make the collection of the data more understandable. 

Response: Thank you for the suggestion. Information on self-assessment of dietary and physical activity has been provided. Kindly see lines 257 to 258.

4. The thematic results should be better structured and visualized, for example with the help of figures or tables. 

Response: Thank you for the suggestion. A thematic table is provided. Kindly see lines 435 to 470.

5. Moreover, I would recommend to reduce the examples of illustrating quotes of participants answers, embed them more into the text or to add them to the appendix, if necessary. 

Response: Thank you, for the suggestion. We have reduced the quotes in the manuscript into the supplementary table (S1 Table). 

Discussion 

1. Due to the quantity of findings it could also become clearer to the reader if there would be subheadings. 

Response: Thank you for the suggestion. Subheadings have been provided within the discussion section. Kindly see lines 762 to 912.

2. Would also advise to be careful with statements like in lines 748 to 750, because they are not scientifically provable 

Response: Thank you for the caution. We have rephrased the statements for readers to know it is anecdotal and unproven. Kindly see lines 854 to 864. 

3. In line 752 an exact reference is needed. 

Response: Thank you for the comment. A reference has been provided. Please, see line 865.

4. In lines 795 it would be more understandable if the authors explain why the results cannot be generalized. Is it due to the sample size and/or cultural and other differences between African and, e.g., European countries, etc? 

Response: Thank you for the suggestion. More clarity has been provided on why we indicated results cannot be generalized. Kindly see line 914 to 918. 

5. In addition, interviews may be subject to bias (e.g. socially desirable responses) because only the subjective experience is recorded. 

Response: Thank you for the suggestion. A statement on this limitation (see lines 917 to 918) and in addition a strength (see line 924 to 925) of the study were provided. 

Conclusion 

1. In the abstract, there is mention of the need for a program to support trainees. However, this information is missing from the conclusion section of the manuscript. Response: Thank you for this observation. The manuscript conclusion has been corrected to reflect that of the abstract. Please see lines 933 to 935. 

2. Also missing is information about new research questions and further studies that should be conducted building on these findings. Overall, the conclusion lacks the bottom line of what we can draw from the study results. 

Response: Thank you for this observation. A new research question and area of research have been added to the conclusion section. Kindly refer to lines 935 to 939.

---

## [Editor Report · Decision Letter 1]

9 Jun 2022

Promoting healthy lifestyles among nurse trainees: perceptions on enablers and barriers to dietary and physical activity behaviours

PONE-D-22-03540R1

Dear Dr. Doegah,

We’re pleased to inform you that your manuscript has been judged scientifically suitable for publication and will be formally accepted for publication once it meets all outstanding technical requirements.

Final editor comment. The manuscript has improved much as you were able to address all the points raised by the referee. It is also, in my view, an interesting work on a highly relevant topic. Even you qualitative approach is appreciable as there are aspects of knowledge that can only be studied using a hermeneutic approach. There are only some minor points that can be adjusted after acceptance. This includes: some typos, check reference information for completeness, tables might be presented in APA norm.

Kind regards,

Michael B. Steinborn, PhD

Section Editor

PLOS ONE
---

## [Editor Report · Acceptance letter]

14 Jun 2022

PONE-D-22-03540R1 

Promoting healthy lifestyles among nurse trainees: perceptions on enablers and barriers to dietary and physical activity behaviours 

Dear Dr. Doegah:

I'm pleased to inform you that your manuscript has been deemed suitable for publication in PLOS ONE. Congratulations! Your manuscript is now with our production department. 

Kind regards, 

on behalf of

Dr. Michael B. Steinborn 

Section Editor

PLOS ONE